# LEARNING FOURIER-SPARSE FUNCTIONS ON DAGS

**Bastian Seifert**,[*] **Chris Wendler**[*] **and Markus Püschel**
Department of Computer Science, ETH Zurich
{baseifert,wendlerc,markusp}@ethz.ch

## ABSTRACT

We show that the classical Moebius transform from combinatorics can be interpreted as a causal form of Fourier transform on directed acyclic graphs (DAGs). The associated Fourier basis, which is spanned by the columns of the zeta transform, enables us to make use of Fourier-sparse learning methods to learn functions on the vertices of DAGs from few observations. As a prototypical application example we construct a DAG from a dynamic contact tracing network, in which each vertex represents an individual at a given timestamp, and learn the function that indicates which of the vertices are infected by a disease.

## 1 INTRODUCTION

The classical Moebius transform (Rota, 1964) plays a key role in enumeration problems and other applications in combinatorics (Stanley, 1986). It can be viewed as a difference operator on a partially ordered set (poset) that, in turn, is equivalent to a transitively closed directed acyclic graph (DAG). DAGs encode causal relationships between events (the nodes) and we are concerned with data associated with the nodes of a DAG.

In this work, we first show that the Moebius transform associated with a DAG can be interpreted as a Fourier transform. Namely, we use the general theory in (Püschel & Moura, 2006) to derive a space of linear convolution operators that are jointly diagonalized by the Moebius transform. Just like for the standard convolution, each such operator is a linear combination of shifts, but the shifts are causal in a sense that we explain. In particular, the value associated with a node depends only on all predecessors. Further, all DAGs with the same transitive closure, i.e., the same dependencies among events, are indistinguishable in our framework.

Our approach is non-stochastic and thus different from the stochastic theory of causal machine learning (Schölkopf, 2019), where the random variables associated with the nodes of a causal DAG only depend on their parents. Rather, our contribution falls into the general research direction of learning on non-Euclidean domains based on convolutions and Fourier transforms (Bronstein et al., 2017). Generalized notions of convolution (Shuman et al., 2013; Sandryhaila & Moura, 2013) laid the foundation of the now celebrated graph convolutional neural networks (Bruna et al., 2014; Kipf & Welling, 2016) and generalized Fourier transforms led to various sample efficient learning algorithms for functions that are sparse in the associated Fourier bases: Hassanieh et al. (2012) devised an algorithm for learning Fourier-sparse time signals, Amrollahi et al. (2019) adapted the same algorithm to the powerset domain, where also Stobbe & Krause (2012) and Wendler et al. (2021) made use of Fourier-sparsity for learning set functions. More recently, Fourier basis vectors also have been used to devise positional and structural encodings for transformers (Vaswani et al., 2017; Kreuzer et al., 2021) and to improve graph neural networks (Dwivedi et al., 2021). Additionally, our novel Fourier basis given by the columns of the zeta transform spans the substructure poset feature space that has been used for learning on structured data such as graphs in (Nowozin, 2009).

In this work, we focus on learning Fourier-sparse functions on the vertices of DAGs using a novel form of convolution, Fourier transform, and associated Fourier basis, different from prior work on general graphs. As a preliminary application example, we consider a dynamic contact tracing network from Kissler et al. (2018) with associated infection signal. The contact tracing network gives rise to a DAG (Kim & Anderson, 2012) that has a vertex for every individual at every given timestamp and two vertices are connected from time $t$ to time $t + 1$ if the corresponding individuals

---

[*]Equal contributions.

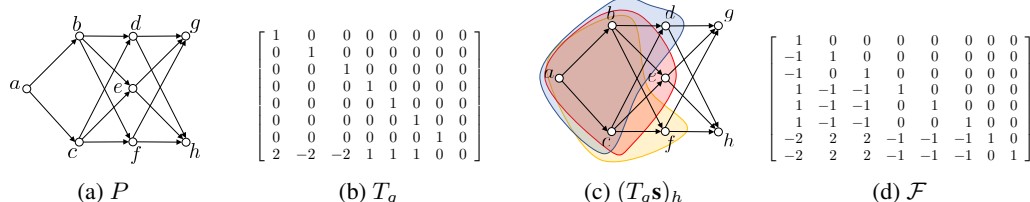

(a) $P$        (b) $T_g$        (c) $(T_g \mathbf{s})_h$        (d) $\mathcal{F}$

Figure 1: Cover graph of poset $P$, causal shift $T_e$, interpretation of causal shift $T_e$, and causal Fourier transform $\mathcal{F}$ of $P$. The rows and columns of $T_e$ and $\mathcal{F}$ are indexed by the poset elements in alphabetical order.

were in close proximity at time $t$. Given this DAG, we simulate the spread of a disease using the model in Kissler et al. (2018) and learn the function that indicates which of the vertices correspond to infected individuals from few samples. In this experiment, our causal Fourier basis associated with the DAG significantly outperforms other graph Fourier bases from (Shuman et al., 2013; Sandryhaila & Moura, 2013) based on the undirected adjacency and Laplacian matrix of the DAG.

## 2    CAUSAL SHIFT, CONVOLUTION, AND FOURIER TRANSFORM ON DAGS

In this section we introduce a causal model for signals, or functions, on DAGs, drawing motivation from signal processing. Following the general theory and approach of algebraic signal processing (Püschel & Moura, 2006; 2008) we obtain a Fourier basis by diagonalizing an appropriate notion of shift and associated convolution. We start with the needed background on partially ordered sets and DAGs.

**Partial orders.** A partially ordered set (poset) is a set $P$ (here we assume finite) with a partial order $\leq$, which is a binary relation that satisfies for all $a, b, c \in P$

1. $a \leq a$ (reflexivity),
2. $a \leq b$ and $b \leq a$ implies $a = b$ (antisymmetry),
3. $a \leq b$ and $b \leq c$ implies $a \leq c$ (transitivity).

If $a \leq b$ and $a \neq b$ we write $a < b$.

**Cover graph.** For $a, b \in P$, the element $b$ *covers* $a$, if $a < b$ and there is no element in between. The cover graph of the poset $P$ is $D = (P, E)$, with edges $E = \{(a, b) \mid b \text{ covers } a\}$. We draw the cover graph from left (smaller elements) to right (larger elements).

**Relation between DAGs and Posets.** The cover graph $D$ of a poset is a directed acyclic graph (DAG). Conversely, every DAG $D = (V, E)$ defines a poset $V$ via the following partial order relation: for $x, y \in V$, $y \leq x$ if $y$ is a predecessor of $x$, i.e., if there is a path from $y$ to $x$. The cover graph of this poset is the transitive reduction of $D$, i.e., the DAG with the fewest number of edges that defines the same poset.

**Events and causes.** We assume a given DAG $D = (V, E)$ and its induced partial order $\leq$. Since all following definitions only rely on $\leq$, we assume without loss of generality that $D$ is in transitively reduced form. We consider the nodes $V$ of $D$ as events and say that event $y$ is a cause of event $x$ if $y \leq x$.

**A causal signal model.** We assume that with every event $y \in V$ there is an associated, unknown cost (or reward) $c_y \in \mathbb{R}$. Further, let $\mathbf{s} = (s_x)_{x \in V}$ be an observed signal on $D$ and assume

$$s_x = \sum_{y \leq x} c_y, \quad \text{for } x \in V. \tag{1}$$

By (slight) abuse of notation we also say that $c_y$, for $y \leq x$, is a cause of $s_x$. $\mathbf{s}$ is causal since each $s_x$ only depends on its predecessor nodes.

E.g., assume a river network that is polluted (with intensity $c_y$) at various (a finite set of) locations. Measuring the pollution at each such location $x$ yields the cumulated pollution $s_x$ from upstream.

**Calculating the causes.** An immediate question is whether the causes $\mathbf{c} = (c_y)_{y \in V}$ can be computed from the observed signal $\mathbf{s}$. This is indeed possible using the well-known Moebius inversion

formula from combinatorics (Rota, 1964). Applied to our setting, it states that

$$s_x = \sum_{y \leq x} c_y \quad \text{if and only if} \quad c_y = \sum_{x \leq y} \mu(x, y) s_x, \tag{2}$$

where $\mu$ is the Moebius function, which can be computed recursively via

$$\mu(x, x) = 1, \text{ for } x \in V, \quad \text{and} \quad \mu(x, y) = - \sum_{x \leq z < y} \mu(x, z), \text{ for } x \neq y. \tag{3}$$

Next, we show that the Moebius transform in (2) can be viewed as a kind of Fourier transform by providing an intuitive notion of shift and associated convolution that it diagonalizes. To our best knowledge this notion of shift and interpretation of the Moebius transform on posets is novel.

**A causal shift.** Given a signal **s** on $D$ as in (1), we define for every $q \in V$ an associated linear shift operator as

$$(T_q \mathbf{s})_x = \sum_{y \leq x \text{ and } y \leq q} c_y, \quad \text{for all } x \in V. \tag{4}$$

In words, the result is the sum of all common causes of $q$ and $x$. Replacing $c_y$ by the formula in (2), we see that $T_q$ is a linear operator on **s**:

$$(T_q \mathbf{s})_x = \sum_{y \leq x \text{ and } y \leq q} \sum_{z \leq y} \mu(z, y) s_z. \tag{5}$$

In Fig. 1b we show the matrix representation of one such shift $T_g$. Like most causal shifts, $T_q$ is not invertible. Note that, if there exists an unique maximum its corresponding shift is invertible.

Fig. 1c illustrates (5) with an example. $(T_g \mathbf{s})_h$ sums all joint causes of $g$ and $h$, which lie in the union of the three colored areas. To obtain this sum, we add $s_d, s_e$, and $s_f$, which however triple counts the causes in the intersection. Thus, one has to subtract two times $s_b$ and $s_c$ and then add $2s_a$:

$$(T_g \mathbf{s})_h = s_f + s_e + s_d - 2s_c - 2s_b + 2s_a = c_f + c_e + c_d + c_c + c_b + c_a. \tag{6}$$

Note that there is not one shift that generates all others (as with the standard translation in 1D). However, as we show below, the Moebius transform diagonalizes all $T_q$ simultaneously (which is possible since *all shifts commute* due to (4)).

We also note that a shift can be viewed as a kind of delay of the set of predecessor nodes of $x$, which gets reduced by shifting.

**Filters and convolution.** Filters are linear combinations of shifts (Püschel & Moura, 2006; 2008), and provide a notion of convolution. In our case, for $\mathbf{h} \in \mathbb{R}^{|V|}$ convolution becomes

$$\mathbf{h} * \mathbf{s} = \Big( \sum_{q \in V} h_q T_q \Big) \mathbf{s}. \tag{7}$$

It also follows that filters are shift-equivariant w.r.t. all shifts, i.e., $\mathbf{h} * T_q \mathbf{s} = T_q(\mathbf{h} * \mathbf{s})$ for all $q \in V$.

**Fourier basis and frequency response.** We now derive the Fourier basis, which are signals that are simultaneous eigenvectors of all shifts. We denote with $\iota_{\{y \leq x\}}$ the characteristic function of $y \leq x$, that is

$$\iota_{\{y \leq x\}} = \iota_{\{y \leq x\}}(x, y) = \begin{cases} 1, & y \leq x, \\ 0, & \text{else.} \end{cases} \tag{8}$$

From the Moebius inversion formula (Rota, 1964) the following theorem follows.

**Theorem 1** (Fourier basis). *For any $y \in V$ the vector*

$$\mathbf{f}^y = (f_x^y)_{x \in V} = (\iota_{\{y \leq x\}})_{x \in V} \tag{9}$$

*is a simultaneous eigenvector of all shift matrices $T_q$, $q \in V$.*

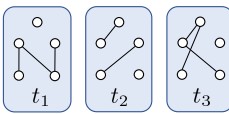 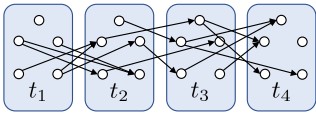 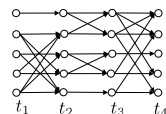

(a) A dynamic network $(V, E_t)$, where the edges change with time $t = t_1, t_2, t_3$.

(b) Copied graphs with new directed edges. The edges $(u, t) \rightarrow (u, t+1)$ are not yet included.

(c) The final DAG $D = (V', E')$; the nodes at each time step are drawn vertically aligned.

Figure 2: Constructing a DAG from a dynamic network (Kim & Anderson, 2012).

*Proof.* According to Rota (1964) the Moebius inversion formula (2) states that the Moebius function and $\iota_{\{y \leq x\}}$ are related via $\sum_{x \leq z \leq y} \mu(x, z) \iota_{\{y \leq z\}}(z, y) = \delta_{\{x, y\}}$, where $\delta_{x, y} = 1$ if $x = y$ and $\delta_{x, y} = 0$ otherwise. Thus it follows that

$$(T_q \mathbf{f}^y)_p = \sum_{\substack{x \leq p \\ x \leq q}} \sum_{z \leq x} \mu(x, z) f_z^y = \sum_{\substack{x \leq p \\ x \leq q}} \delta_{\{x, y\}} = \begin{cases} 1 & \text{if } y \leq p \text{ and } y \leq q, \\ 0 & \text{else.} \end{cases}$$

This means that $T_q \mathbf{f}^y = 1 \cdot \mathbf{f}^y$ if $y \leq q$, and $T_q \mathbf{f}^y = 0 \cdot \mathbf{f}^y$ otherwise. $\qquad\square$

**Fourier transform.** The previous discussion shows that $\mathbf{s}$ and $\mathbf{c}$ form a Fourier transform pair in our model, i.e., $\mathbf{c} = \widehat{\mathbf{s}}$. Namely, the Fourier expansion $\mathbf{s} = \sum_{y \in V} \widehat{s}_y \mathbf{f}^y$ implies $s_x = \sum_{y \in V} \iota_{\{y \leq x\}} \widehat{s}_y = \sum_{y \leq x} \widehat{s}_y$, which is exactly (2) and inverted as shown there. Thus the Fourier transform in matrix form becomes $\widehat{\mathbf{s}} = \mathbf{c} = \mathcal{F}_D \mathbf{s}$, with $\mathcal{F}_D = [\mu(x, y) \iota_{\{x \leq y\}}]_{y, x \in V}$.

## 3 INFECTION SIGNALS ON DYNAMIC DAGs

In many situations, networks (i.e., graphs) are non-static. For example the proximity of persons, important for contact-tracing of infectious people during a pandemic, changes with time leading to a dynamic network structure. Other examples include peer-to-peer networks between vehicles in traffic or transactions between traders in a market. Using the method proposed in Kim & Anderson (2012) we turn the contact tracing network from Kissler et al. (2018) into a DAG. Then we describe an infection model, which yields signals on the contact tracing DAG.

**The Haslemere data set.** In Kissler et al. (2018) a dynamic network based on smartphone proximity was reported and used to model the spread of a disease. The dynamic network consists of $|V| = 469$ participants whose pairwise distance was measured for three days every 5 minutes between 7am and 11pm using a smartphone app, leading to 576 timestamps, each with a complete graph. We remove all edges with distance $> 20$ meters due to the infection model below.

**Dynamic DAGs.** The Haslemere network is dynamic, i.e., a collection of graphs $G_t = (V, E_t)$ where the set of edges $E_t$ changes with time $t \in T = \{t_1, \ldots, t_m\}$. We create a DAG $D = (V', E')$ from the data as follows. We add an additional timestamp $t_{m+1}$ and make a copy of the node set for each timestamp, i.e., the new node set is $V' = \{(v, t) \mid v \in V, t \in T \cup \{t_{m+1}\}\}$. Further, we connect nodes $(u, t)$ with $(v, t+1)$ if $(u, v) \in E_t$ and always $(u, t)$ with $(u, t+1)$ to form $E'$. The construction is illustrated on a small example in Fig. 2. Since we want to compare our work to prior graph Fourier transforms, which require eigendecompositions of Laplacian or adjacency matrix, we have to restrain the size of the graph. Hence, we only consider the contact data every hour, resulting in $|T| = 37$ timestamps and a DAG with $|V'| = 17612$ nodes and $|E'| = 24596$ edges.

**Infection signals.** The susceptible-exposed-infectious (SEI) model in Kissler et al. (2018) simulates the spread of a disease from a number of initially infected individuals. In the SEI model a healthy individual becomes infected with a certain probability when exposed to an infectious individual. As exposition time we choose one hour, matching our chosen time resolution.

## 4 LEARNING FOURIER-SPARSE CAUSAL SIGNALS

We use the data and model in Section 3 to generate DAG signals. Concretely, we start with $k = 2, 5, 9$ individuals infected at random at time $t_1$ and propagate infections probabilistically as

described in the SEI model. For each $k$ we ran the simulation ten times, resulting in 30 DAG signals. The signals have value 1 at node $x = (u, t)$ if the individual $u$ is infected at time $t$ and 0 otherwise.

Our goal is to reconstruct the signal **s** from few samples. In our setting this means testing random individuals at random time points and trying to determine who got infected and at what time. To do so, we use the Fourier basis vectors as features for a sparse linear binary classifier (Ng, 2004). We then apply this approach with our Fourier basis (Theorem. 1), and, for comparison, with prior graph Fourier bases obtained from the undirected versions of the adjacency matrix and Laplacian of the DAG.

**Fourier-sparse learning.** The basic idea is to approximate a binary signal **s** with $\tau(\sigma(\mathbf{r}))$, where **r** is Fourier-sparse, $\sigma$ is the logistic sigmoid function ($\sigma(x) = 1/(1 + \mathrm{e}^{-x})$), applied elementwise, that converts real values to probabilities, and $\tau$ is a threshold function. In our case, $\tau$ simply rounds elementwise to 0 or 1. Formally, the probability that a node $x \in V$ is infected is $p(x) = \mathrm{prob}(x \text{ is infected}) = \sigma\left( \sum_{y \in V} \widehat{r}_y f_x^y \right)$, where $\widehat{r}_y = 0$ for most $y \in V$.

Assume $n$ observed signal values $s_1, \ldots, s_n$ at random nodes $x_1, \ldots, x_n$, respectively. The non-zero Fourier coefficients $\widehat{r}_y$ are estimated by solving a logistic regression problem, regularized by an $L^1$-loss term to ensure Fourier-sparsity of **r** (Ng, 2004). The optimization problem is given as

$$\min_{\widehat{\mathbf{r}} \in \mathbb{R}^{|V|}} - \sum_{i=1}^{n} s_i \log p(x_i) + (1 - s_i) \log(1 - p(x_i)) + \lambda \sum_{y \in V} |\widehat{r}_y|, \qquad (10)$$

where $\lambda \ll 1$ is a hyperparameter. We found that $\lambda = 0.1$ worked well for all bases.

**Experiment and results.** We observe a fraction $n/|V|$ of the signal and then use (10), with three different Fourier bases, to obtain $\widehat{\mathbf{r}}$, which in turn determines **r** and thus the prediction $\tau(\sigma(\mathbf{r}))$ of **s**.

Fig. 3 shows the accuracy $\mathrm{acc}(\mathbf{r}) = |\{x \in V \mid s_x = \tau(\sigma(r_x))\}|/|V|$ of the prediction as function of the fraction of values observed. The solid lines show the mean and the shaded area the 95% confidence interval over the 30 different signals. We include the trivial estimates "all infected" and "nobody infected," i.e., **s** is constant 1 or constant 0, respectively, as dotted lines.

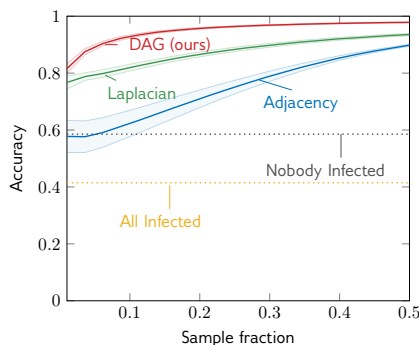

Figure 3: Results of the proposed learning approach on the Haslemere signals.

The estimation based on our proposed causal Fourier basis performs best, beating both graph Fourier bases associated with the Laplacian and adjacency matrix. We note that in our causal Fourier domain the signals are indeed sparse: computing the full spectrum of dimension 17612 shows about 90% sparsity. It shows that for the considered signal the combinatorial causal Fourier basis yields a better representation than the more geometric prior graph Fourier bases.

## 5    CONCLUSION

Our main contribution is a novel, mathematically sound form of convolution and Fourier analysis for causal data on DAGs, giving a new interpretation to the classical Moebius transform from combinatorics. Our framework is fundamentally different from prior graph Fourier analysis as the associated shifts, which the Fourier transform diagonalizes, capture causal dependence and not proximity like the adjacency or Laplacian matrix. Furthermore, for DAGs the classical graph Fourier bases do not exist, as neither adjacency matrix nor Laplacian are diagonalizable for DAGs. Our prototypical application example shows the potential of Fourier-sparse learning within our framework. To further broaden the applicability it will be important to extend the theory to weighted DAGs to model a broader set of applications.

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
