# OpenReview forum: "Learning Fourier-Sparse Functions on DAGs"
_ICLR.cc/2022/Workshop/OSC — ICLR2022 OSC  Oral_

### Official Review · Reviewer_SLjw · 2022-03-10
**Interesting idea, elegant paper**

**Rating:** 3
**Confidence:** 3

**Review:**

The paper discusses Fourier transformations of signals on partial orders and uses these to define convolutions on such signals. For partial orders, a signal $c$ causes a signal $s$ when one sums the value of the predecessors. This is a bijection, whose inverse is the Möbius transform. For any node $q$, it defines a linear "shift" transformation on signals whose value at node $x$ is the sum of values at nodes that are predecessors to both $x$ and $q$. These shift operators commute for all nodes $q,q'$ and can thus be simultaneously diagonalized, giving the Fourier basis.  It turns out the the previously mentioned signal and its cause are exactly related by a Fourier transform.

The authors propose to predict missing signal values by fitting a sparse Fourier spectrum on a partially observed signal. Hereby the proposed basis outperforms a Laplacian or adjacency basis.

I like this paper as it proposes a Fourier transform on a novel data type and elegantly shows how this outperforms other bases. Also, it is very clearly written and a joy to read.
I have two suggestions for points for improvement:
1) why is the proposed method formulated as if acting on signals on DAGs, while it seems like the method is only affected by the induced partial order. For example, the Fourier basis for a complete DAG would be the same as for the chain? Wouldn't a presentation around partial orders then make more sense?
2) Personally, I would have liked to see additional experimental comparisons to e.g. graph neural networks.

And a question:
An often desired property of convolution operators is that they commute with automorphisms of the structure. This could be implemented by letting the convolution in Fourier space have the same coefficient for each eigenspace. Could the authors maybe comment on whether this is desirable in their context? It seems like there would be only two eigenvalues and thus only two parameters? How does this contrast with the graph Laplacian?

---

### Official Review · Reviewer_PFWS · 2022-03-14
**Fourier transforms on DAGs with application to contact tracing**

**Rating:** 3
**Confidence:** 3

**Review:**

The paper presents an analysis of a generalized Fourier transform for signals on DAGs. This is a linear transformation that diagonalizes a generalized form of "causal shift" on DAGs. The Fourier features are used to classify infection status in a contact tracing application.

I found the paper mathematically interesting, innovative and elegant. After reading I was left with a few questions which could be answered in the text (see below). I also wonder how well the mathematical setup fits the application (again see below). Nevertheless I think this is a strong workshop submission, with fundamental ideas about Fourier transforms on graphs, that have potential to impact future work in various areas.

Detailed comments:

Some technical questions that could be answered in the text:
- When is the shift operator invertible?
- Do shifts commute?
- How do we see that a shift operator is diagonalizable? How do we see that all shifts are simultaneously diagonalizable? (This would follow if shifts always commute)

In the sentence: ".. filters are shift-invariant (resp. equivariant)". I don't understand the word respectively here, as only one noun is mentioned before (filters). The equation that follows shows equivariance not invariance.

It would be good to explain the intuition of the contact tracing model and Fourier features a bit more. As I understand it, the probability that x is infected is modelled as a sparse linear combination of Fourier features. This means that only nodes upstream of x can contribute to its probability of being infected, which makes sense as x can only have been infected through a path of contacts to an infected individual. So a positive coefficient hat{r_y} means that this individual was probably infected, and thus contributes positively to all downstream nodes. However the distance to the downstream node is not taken into account, I think. I don't really understand the interpretation of negative coefficients either: if we are very sure someone is not infected, that could lower the probability that downstream nodes are infected, but not if those nodes have other ancestors that are infected. That is, a negative contribution should not cancel a positive one; addition of reals is probably not the right way to combine contributions. Perhaps you can model this situation nicely with enriched categories / preorders -- see the book on Applied Category Theory by Fong & Spivak.

---

### Decision · Program_Chairs · 2022-03-23

**Decision:**

Accept (Oral)

**Comment:**

Strong paper that presents a novel form of Fourier analysis for causally structured data. Reviewers agree on the novelty, quality of presentation, and value for the workshop. Accepted for an oral spotlight presentation.